# Metabolic Disturbance of High-Saturated Fatty Acid Diet in Cognitive Preservation

**DOI:** 10.3390/ijms24098042

**Published:** 2023-04-28

**Authors:** Antonio Rivas-Domínguez, Himan Mohamed-Mohamed, Margarita Jimenez-Palomares, Victoria García-Morales, Laura Martinez-Lopez, Manuel Luis Orta, Juan José Ramos-Rodriguez, Beatriz Bermudez-Pulgarin

**Affiliations:** 1Department of Cellular Biology, University of Seville, 41009 Seville, Spain; 2Department of Physiology, Faculty of Health Sciences (Ceuta), University of Granada, 51001 Ceuta, Spain; 3Department of Biomedicine, Biotechnology and Public Health, University of Cádiz, 11003 Cádiz, Spain

**Keywords:** dementia, cognition, episodic memory hyperlipidemia, insulin resistance, LDL, saturated fatty acid, HFD, *LDLR*

## Abstract

Aging continues to be the main cause of the development of Alzheimer’s, although it has been described that certain chronic inflammatory pathologies can negatively influence the progress of dementia, including obesity and hyperlipidemia. In this sense, previous studies have shown a relationship between low-density lipoprotein receptor (LDLR) and the amyloid-beta (Aβ) binding activity, one of the main neuropathological features of Alzheimer’s disease (AD). LDLR is involved in several processes, including lipid transport, regulation of inflammatory response and lipid metabolism. From this perspective, *LDLR*^−/−^ mice are a widely accepted animal model for the study of pathologies associated with alterations in lipid metabolism, such as familial hypercholesterolemia, cardiovascular diseases, metabolic syndrome, or early cognitive decline. In this context, we induced hyperlipidemia in *LDLR*^−/−^ mice after feeding with a high-saturated fatty acid diet (HFD) for 44 weeks. *LDLR*^−/−^-HFD mice exhibited obesity, hypertriglyceridemia, higher glucose levels, and early hepatic steatosis. In addition, HFD increased plasmatic APOE and ubiquitin 60S levels. These proteins are related to neuronal integrity and health maintenance. In agreement, we detected mild cognitive dysfunctions in mice fed with HFD, whereas *LDLR*^−/−^-HFD mice showed a more severe and evident affectation. Our data suggest central nervous system dysfunction is associated with a well-established metabolic syndrome. As a late consequence, metabolic syndrome boots many behavioral and pathological alterations recognized in dementia, supporting that the control of metabolic parameters could improve cognitive preservation and prognosis.

## 1. Introduction

In the last decades, obesity has become a significant problem worldwide, i.e., 17–18% of European adults are obese [1,2,3]. Although the etiology is still not fully understood, some risk factors are involved, such as genetics, culture and economic factors, late maternity age, endocrine disruptors, or medicament consumption between others [4].

The excessive fat accumulation in adipose tissue is involved in the secretion of hormones, making obesity a chronic and relapsing disease [5]. High lipid plasma concentrations affect homeostasis at several levels, so lipotoxicity has recently been adopted [6]. Obesity links with insulin resistance, vascular damage, and cognitive decline. Furthermore, it has been considered a risk factor for Alzheimer’s (AD) and Vascular Dementia (VaD) [7,8,9]. The underlying mechanism involves a chronic inflammatory process affecting the Central Nervous System (CNS) [10]. Indeed, an impaired blood-brain barrier (BBB) permeability allows free passage of proinflammatory mediators and fatty acids, the latter related to hypothalamic insulin resistance [11].

Another critical issue is that obese people use to present hypoperfusion at CNS [12], which, together with chronic inflammation, activates neurodegenerative processes [13]. Consequently, clinical manifestations of a progressive decline in memory, rational thinking and learning have been reported [9,14].

Besides, high cholesterol levels are responsible for many clinical manifestations of obesity (for a review, see [15]). Under normal conditions, synaptogenesis or myelin synthesis requires high cholesterol levels [16,17]. In this context, apolipoprotein E (APOE) transports cholesterol to the CNS [18].

Altered APOE levels are related to atheroma plaque formation, amyloidogenesis and cholinergic system impairment [19]. Additionally, APOE is part of low-density lipoproteins (LDL) [19] and interacts with LDL receptors (LDLR). Mutations in the *LDLR* gene cause familial hypercholesterolemia [20,21]. *LDLR* expression is widely distributed in the CNS in neurons, astrocytes, and oligodendrocytes [22]. Indeed, LDLR is involved in synaptic plasticity and neuronal proliferation [23].

The relationship between LDLR and brain inflammation is complex: while an LDLR overexpression reduces inflammation during infections [24], increased amyloid-beta (Aβ) deposits and reduced inflammation have been reported in *LDLR* knockout mice [25]. Moreover, LDLR regulates APOE levels and Aβ deposition in the brain, risk factors for AD [25].

Altogether, these data support a link between LDLR and brain homeostasis. However, the exact implication of LDLR in cognitive decline is not completely dilucidated. In the present paper, we hypothesize that a hypercaloric diet and a deficiency in LDLR can influence early cognitive functions.

## 2. Results

This paper used a similar number of male and female animals for all conditions (Table 1). Nevertheless, no gender differences were found for any analysis performed according to T test (two tails). For this reason, male and female data were pooled at each experimental point.

### 2.1. Body Weight, Triglycerides Levels and Liver Weight

Previous studies in rodents have shown that HFD or hypercaloric diets, maintained for eight weeks, induce a significant increase in body weight, which correlates well with the size of lipid droplets in adipocytes [26].

In good agreement, we found that HFD induced a statistically significant increase in body weight for WT or *LDLR*^−/−^ as compared with RD-fed mice from the 8th to 44th week, while no differences between groups were found in the first weeks of diet (Figure 1A). From the 19th week, the body weight of WT mice fed with HFD was statistically different (*p* = 0.024) to all the RD groups [week 19: F (3, 37) = 3.536 † *p* < 0.01 Wt-Butter versus Control and LDLR^−/−^-RD groups].

From the 20th week, all groups fed with HFD presented statistically different body weights than RD groups (Week 20: *p* = 0.003; and Week 21 to Week 44: *p* < 0.001).

As expected and consistent with previous studies, the HFD in our mouse model raised triglyceride levels in the *LDLR*^−/−^ animals (Figure 1B).

According to previous literature, a hypercaloric diet has been related to hepatomegaly. In relation to this, *LDLR*^−/−^ mice present increased levels of hepatic lipid droplets due to a hypercholesteremic diet [27,28]. In our hands, HFD-fed mice showed increased liver weights as compared to RD groups [F (3, 36) = 21.736; ** *p* < 0.001]) (Figure 1C).

### 2.2. Glucose Levels and Pancreatic Mass Weight

Being overweight and an excess fat intake are serious risk factors for developing type 2 diabetes mellitus, a disease characterized by an excess of fasting glucose. Indeed prolonged HFD diets induce a pro-inflammatory state that finally affects signaling pathways downstream membrane insulin receptors [29,30], leading to reduced glucose tolerance despite elevated insulin levels, both of them good indicators of insulin resistance (For a review, see [31]).

Previous reports have shown that chronic HFD for 13–22 weeks induces weight gain, increased glucose levels and insulin resistance [32,33]. In agreement, our data (Figure 2) shows that continuous HFD from the 20th to the 44th week increased the levels of plasmatic glucose in *LDLR^−^^/−^* mice compared with the rest of the groups. Basal levels of plasmatic glucose showed no differences [F (3, 36) = 1.084; *p* = 0.368], but 12 weeks of HFD (20th week) induced augmented glucose levels in mutant mice as compared with the rest of the experimental groups [F (3, 32) = 11.62; ** *p* < 0.001 versus rest of the groups]. On the other hand, glucose analysis in the 32nd and 44th week showed significant differences between *LDLR^−^^/−^* mice fed with HFD and Control groups [F (3, 31) = 4.082; ^†^ *p* = 0.015 and F (3, 29) = 3.119; ^†^ *p* = 0.041, respectively] (Figure 2A).

In consonance with the data presented above, mutant mice fed with HFD showed hypertrophic pancreas (Figure 2B) as compared with those raised with RD or Control [F (3, 37) = 4.796; ^††^ *p* < 0.006].

### 2.3. Cognitive Task: NOD Test

In the present paper, we aimed to analyze a possible role of HFD and/or a defective clearance of plasmatic LDL in cognitive functions. For that, the NOD test for episodic memory was employed. This method has been tested as a sensitive tool for detecting early memory dysfunctions in animal models [34].

Previous reports showed that WT mice supplied with HFD didn’t present cognitive impairment [32,35]. Surprisingly, HFD improved working memory and reduced anxiety in C57BL/6J mice showing that hypercaloric diets might be beneficial in a restricted context [35]. According to Yoshizaki K. et al., the seven-week hypercaloric diet-induced hyperlocomotion suggests a lower state of anxiety and a better resolution of the maze-Y test. However, these data must be taken with caution since the period under HFD was very limited (7 weeks), and the animals were much younger (15 weeks) compared to those we have used in the present study (36 weeks under HFD and 44 weeks of age). More in-depth studies are required to decipher the roles of hypercaloric diets in cognitive functions, especially for episodic memory, usually altered in the first stages of cognitive decline [36,37]. We have evaluated episodic memory in our experimental model to address that question. Total time spent by the objects as well as the paradigms “what”, “where”, and “when”, were evaluated (Figure 3).

Regarding the former (Figure 2A), our data clearly shows that the Control group spent more time exploring objects as compared with the rest of the groups [F (3, 34) = 4.657 ** *p* = 0.008]. However, motor function integrity was assessed in all groups before cognition studies. Furthermore, the actimetry test (day 1 of the NOD task) revealed that distance and distribution areas traveled were similar in all the groups supporting unanxiety.

Regarding episodic memory, our data point to clear alterations after hypercaloric chronic diets. Indeed, HFD groups showed significant differences as compared with RD groups for the paradigm “where” [F (3, 36) = 5.534, ^##^
*p* = 0.003) (Figure 3B). Furthermore, differences were found between *LDLR*^−^^/−^-HFD versus the Control group (F (3, 32) = 4.793, ^††^ *p* = 0.007) for the paradigm “when” (Figure 3C). Contrary to expected, no differences were found for the paradigm “what” [F (3, 33) = 1.714 *p* = 0.183] (Figure 3D). Previous studies with hypercaloric diets did not describe memory disruption in wild-type mice, but it should be noted that to the best of our knowledge, studies with HFD as long as ours (36 weeks) have not been carried out. Therefore, it is feasible that the prolongation of the period under HFD could have caused the limited cognitive impairment described here in WT mice.

We demonstrate that HFD interferes with episodic memory, especially in *LDLR*^−/−^ background.

### 2.4. Plasmatic Levels of APOE and Ubiquitin 60S

APOE is a lipidic transporter protein related to sporadic AD development. Altered APOE levels have been connected with the pathogenesis of Aβ, neurofibrillary Tau tangles accumulation, inflammation response by microglia and astrocytes activation and BBB disruption [38]. On the other hand, the ubiquitin system is implicated in many cellular functions, such as proteostasis. In addition, altered ubiquitin function has been related to the most aggressive pathologies forms of AD [39].

In this work, we have monitored plasmatic levels of both proteins as markers of possible CNS damage. We found that APOE levels were higher in the mice fed with HFD compared with RD groups [F (3, 8) = 208.916 ** *p* < 0.001], as shown in Figure 4A. A similar profile was observed when analyzing ubiquitin 60 S levels (Figure 4B). In this sense, the highest ubiquitin levels were found in LDLR^−/−^-HFD mice, whereas intermediate values were found in the LDLR^−/−^-RD and WT-HFD as compared with control [F (3, 8) = 1545.947 ** *p* < 0.001 versus rest of the groups and ## *p* < 0.001 versus Control and LDLR^−/−^-HFD groups].

## 3. Discussion

Epidemiologic data have shown that metabolic disorders influence fine-tuned cognitive functions. In this sense, metabolic syndrome or hypercholesterolemia are considered risk factors for cognitive decline and AD or VaD dementia [9,40].

Nonetheless, the underlying mechanisms involved have not been fully dilucidated. There are conflicting data about the exact role of hyperlipidaemia in this sense. Normal range plasmatic lipid levels are required for healthy brain physiology. Indeed, alterations in these values have been linked to improper brain tasks, which create the perfect niche for dementia outcomes [41,42].

As expected, HFD induced overweight in our model system as detected from the 19th week, in good agreement with previous studies [43,44]. As a result, HFD-fed animals doubled the weight found in RD mice. High abdominal fat levels, such as those in HFD-fed *LDLR*^−/−^ animals, influenced proper carbohydrate and lipid metabolism. On the contrary, WT-HFD animals showed overweight but no metabolic alterations. Hence, our data point to a synergistic metabolic impairment when a defective LDLR is combined with a hypercaloric diet (Figure 1B and Figure 2).

The high glycaemic index observed in these conditions is likely a consequence of pancreatic fat infiltration [45], which also explains insulin resistance and elevated pancreatic weight in *LDLR*^−/−^-HFD animals. Indeed, hypertriglyceridemia and insulin resistance are typical features of metabolic syndrome, obesity and/or type 2 diabetes mellitus [46].

Our data suggest that a hypercaloric diet is insufficient to promote elevated glycaemic index, but defects in LDLR result in high plasmatic glucose and lipid-related pathologies. Glucose levels between 200 and 300 mg/dL, such as those observed in this paper, have been observed in prediabetes syndrome murine models [43,47]. In this sense, insulin resistance promotes de novo hepatic lipogenesis and consequently increases VLDL and LDL [48,49], much like those found in *LDLR*^−/−^-HFD animals.

Additionally, HFD-fed animals presented increased hepatic weights (Figure 1C). Previous reports showed that hepatomegaly and hepatic steatosis are related to increased levels of *AKT* expression [50] and subsequent lipid and glycogen accumulation [51] which explains hepatic mass augmentation. Furthermore, AKT is essential for insulin metabolism and promotes glucose absorption in adipocytes and lipogenesis [52], explaining an augmented corporal mass.

Most LDL particles enter intracellularly by LDLR-dependent endocytosis in hepatocytes. Our data show increased levels of APOE, a ligand of LDLR, in HFD-fed mice. In speculation, elevated APOE levels result from a negative loop mechanism to reduce plasmatic TG levels.

We believe the excess of plasmatic TG observed when *LDLR*^−/−^ mice are fed with HFD promotes overweight and pancreatic fat deposits. Pancreatic size is correlated with corporal weight, where obese people have fat deposits in the pancreas [53] and liver [54], increasing the risk for insulin resistance and type 2 diabetes [55]. Furthermore, we believe that a high glycaemic index, an early insulin resistance symptom found in *LDLR*^−/−^-HFD mice is a consequence of pancreatic fat deposit accumulation. Altogether, *LDLR*^−/−^-HFD show classic phenotypic patterns for metabolic syndrome.

Type 2 diabetes, obesity and metabolic syndrome are risk factors for VaD or AD, as shown by numerous reports [14,40,56,57]. In addition, impaired episodic and work memory and reduced global cognition levels are also connected with these metabolic alterations [9,58,59].

Episodic memory results are affected in the first stages of dementia when brain atrophy is not detectable [60,61,62]. In the present paper, we have used the NOD task, which allows a rigour exploration of episodic memory. *LDLR*^−/−^-HFD animals showed worse cognitive efficiency than the rest of the groups, in agreement with previous studies [63,64]. However, the brain weights of all groups were similar, with no signs of brain atrophy being detected. We found that a deficiency in LDLR and HFD altered the “where” and “when” paradigms. A plausible explanation for our data is that BBB was damaged by high glucose and TG plasmatic levels, as previously reported [65]. Olivera et al. found that increased permeability at the hippocampal BBB level was related to cognitive decline and plasmatic cholesterol levels [65].

A damaged BBB has been associated with a defective Aβ peptide degradation and its subsequent accumulation. Aβ pathology is considered an early symptom of AD and promotes inflammation and neurodegeneration [66]. In line with this, two months of HFD in *LDLR*^−/−^ mice resulted in elevated levels of Aβ_1–42_ in the brain cortex [67], while five months were enough to detect them by immunoreactivity in rats [68]. In both cases, a slight cognitive decline was observed in line with the data presented here. The molecular mechanisms here remain elusive but likely involve Aβ pathology and neurotoxicity, as previously reported in *LDLR*^−/−^ raised with a hypercaloric diet [69]. However, some studies indicate *LDLR*^−/−^ mice presented suppressed microglial activation in the cortex and hippocampus [63,70]. So, we cannot rule out the possibility that the altered cognitive function observed is likely a consequence of Aβ pathology-induced neurotoxicity rather than inflammation.

AD is also highly influenced by APOE, which relates to Aβ/Tau pathologies and inflammation [71]. Therefore, we believe that elevated plasmatic APOE levels and BBB damage could favour the early development of dementia. Furthermore, we conjecture that vascular damage would impede Aβ elimination through the BBB, which, together with increased APOE levels, increases Aβ brain accumulation. This scenario would cause inflammation and hence cognitive impairment.

In our experimental model, we have observed high plasmatic levels of ubiquitin 60 S. Ubiquitin is a small protein involved in several cellular processes such as protein degradation, DNA repair, antigen processing, cell cycle, etc. Damaged mitochondria are eliminated by mitophagy, a ubiquitin-dependent process where these organelles are finally degraded by lysosomes [72]. Dysfunctional mitophagy is an early pathology involved in AD, which results in an accumulation of damaged mitochondria [73]. Oxidative stress is a consequence of impaired mitophagy [74,75] and contributes to synaptic dysfunction and Aβ/Tau deposits in AD patients [76].

Previous reports showed that functional mitophagy reduced either Aβ or Tau deposit progression and hence the cognitive decline in murine models for AD [39]. On the other hand, physical activity and a healthy diet promote mitochondrial biogenesis and mitophagy and improve synaptic plasticity [77]. In our hands, an HFD or a defective LDLR are sufficient to enable alterations in plasmatic levels of ubiquitin 60 S. However, combined factors induced even more augmented levels of ubiquitin. Additionally, total exploring time (Figure 3A) showed reduced physical activity in HFD and *LDLR*^−/−^ mice, suggesting a possible decrease in mitophagy and mitogenesis even though more research is required in this field.

## 4. Materials and Methods

### 4.1. Animals

This paper employed knockout mice for the low-density lipoprotein receptor (C57/6-*LDLR*^−/−^) and wild-type C57/6. Under a regular diet (Chow diet, RD), *LDLR*^−/−^ mice present increased levels of plasmatic cholesterol due to a defect in the clearance of very low-density lipoprotein (VLDL) and LDL [78]. Unfunctional receptor-coupled endocytosis of VLDL or LDL has been related to atherosclerotic lesions and hypercholesterolemia in *LDLR*^−/−^ mice. The severity of these lesions is even more augmented when knockout animals are fed a high-fat (HFD) or high-cholesterol diet [79]. These features make C57/6-*LDLR*^−/−^ a suitable and well-accepted model for studying human hypercholesterolemia [80,81].

Mice were purchased from Jackson Laboratory. All animals were kept under the following conditions: 23 ± 2 °C, 55 ± 5% relative humidity, 12 h light/dark cycle and ad libitum feeding.

To induce a hypercholesterolemia status, animals were fed from the 8th to 44th weeks with HFD. While RD contains only 3% of fats, these levels climb up to 56% in HFD, of which 53% are milk cream fats. Milk cream fats contain 35% of saturated fatty acids (SFA), mainly palmitic acid, Table 2.

Timeline, the period under diet and the mice’s age are presented in Figure 5. All experimental procedures were approved by the Animal Care and Use Committee of the University of Seville in accordance with the Guidelines for Care and Use of experimental animals (European Commission Directive 86/609/CEE and Spanish Royal Decree 1201/2005).

### 4.2. Metabolic Determinations

The body weight of mice was monitored weekly from the 8th to the 44th week. However, as previously described, postprandial blood glucose levels were determined at the 8th, 20th, 32nd and 44th weeks [82]. Briefly, blood glucose levels were measured from nicked tails using the glucometer Optium Xceed (Abbott, Maidenhead, UK).

### 4.3. Locomotor Activity and Cognitive Test

One week before sacrifice, two endpoints, namely actimetry and new object discrimination (NOD task), were chosen to monitor locomotor activity and episodic memory, respectively, as previously described [82].

For the actimetry test (day 1), mice were kept in a rectangular box (22 cm long × 44 cm width × 50 cm high) where the distance walked by mice in 30 min was scored.

NOD tests were performed the following day (Figure 6). Briefly, for familiarization purposes, mice were exposed for 5 min to two objects that were not used anymore. Then, on day three, each mouse received two sample trials and a test trial, as depicted in Figure 6.

For the first sample trial, a blue ball was set up in the box as depicted. First, mice were placed in the center of the box for 5 min. Then, after 30 min, four new objects were placed (red cones) in each corner of the box for 5 min. After another 30 min, a test trial of two blue balls and two red cones was placed in the box’s corners. Finally, identical figures were set up opposite corners (Figure 6).

All the assays were recorded by a video camera. In addition, videos were used to score the time spent by mice in each object during the test trial.

Episodic memory is the capacity to remember specific events from the past [84]. To study episodic memory, previous studies concluded the importance of answering specific questions such as “what”, “where”, and “when” it happened [34]. For that, at a behavioral level, specific methods have been developed (Table 3).

According to previous reports [32,83], “what” happened can be calculated as the timing difference between old familiar (sample 1) and recent familiar (sample 2) objects. On the other hand, “where” happened can be calculated as the time difference spent between displaced and non-displaced objects. Finally, “when” results from calculating the time difference spent between old non-displaced and the mean of recent familiar displaced objects.

### 4.4. Tissue Processing

At the end of the NOD test, mice were sacrificed by thiobarbital overdose (40 mg/kg). Brain, liver, and pancreas were carefully extracted and weighed from each mouse.

### 4.5. Triglycerides Measurements

For plasma determination, blood was collected from the tail vein into capillary tubes precoated with potassium-EDTA (Sarstedt, Nümbrecht, Germany). Blood triglyceride levels were measured in mg/dl using the HDLc-P kit (SPINREACT S.A.U., Girona, Spain), following the manufacturer’s indications, as previously described [85].

### 4.6. APOE and Ubiquitin 60S

Blood was drawn by cardiac puncture, and serum was separated by low centrifugation. Later, VLDL and LDL were removed using a lipoprotein purification kit (ABIN2345055), and total protein quantification was calculated by the BCA Pierce method before analysis. Quantification and identification of plasmatic proteins were analyzed by BRUKER MALDI-TOF/TOF technology and MaxQuant software v2.1.2.0. At the Proteomics Unit of the Central Research Support Service of Cordoba University (https://www.bruker.com/products/mass-spectrometry-and-separations/maldi-toftof/ultraflextreme.html accessed on 12 December 2022).

### 4.7. Statistical Analysis

The numbers of male and female mice used for statistical analysis were specified in Table 3, except for the statistics of APOE and Ubiquitin 60S levels in plasma, carried out in 3 mice per group (two males and one female). Body weight evolution was analyzed by two-way ANOVA. Glucose levels, organ weight and episodic memory values were analyzed by one-way ANOVA for independent samples followed by the Tuckey b test. SPSS v.15 software was used for all statistical analysis.

## 5. Conclusions

Our data suggest that an altered lipid metabolism promoted by either an absence of the LDL receptor or/and a high saturated fat diet is sufficient to initiate a global metabolic imbalance with cognitive implications. These disorders are aggravated when both conditions are concomitant (*LDLR*^−/−^ and HFD). We found high plasmatic fatty acids increase, promoting a massive accumulation of body fat, particularly prominent in the pancreas and liver. We hypothesize that saturated fatty acid accumulation might promote glycaemic dysfunction, compatible with prediabetes. Taken together, our current model describes that the metabolic imbalance observed promotes episodic memory impairment, one of the first observable clinical evidence in dementia through a possible mismatch in mitophagy activity. Our data support moderate saturated fatty acid consumption is crucial for chronic proinflammatory disease prevention and a healthy cognitive state.

## Figures and Tables

**Figure 1 ijms-24-08042-f001:**
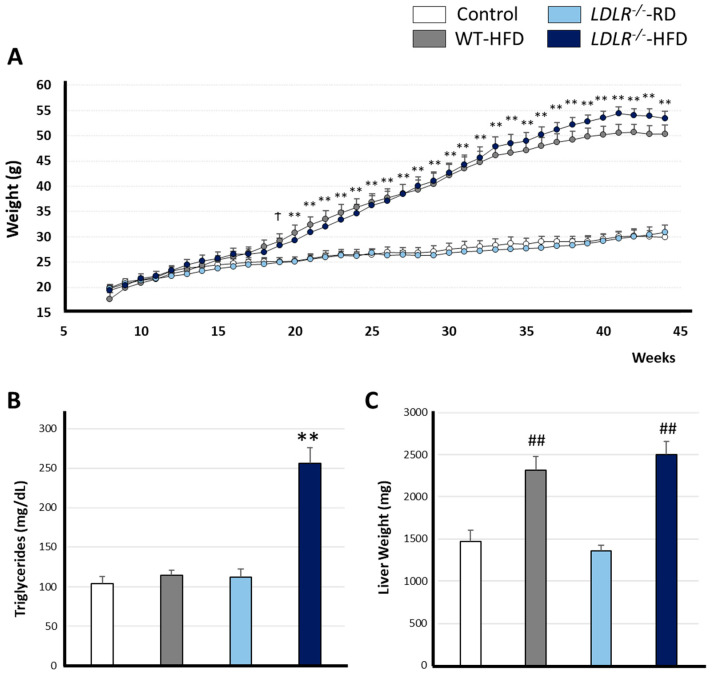
HFD affects body weight, triglyceride levels and liver mass. (**A**) Body weight progression from the 8th to the 44th week. HFD induced increased levels of body weight as compared to RD groups. Data show the mean ± standard error from 9–12 mice per group. Differences were considered statistically significant where ^†^ *p* < 0.05 *LDLR*^−/−^-HFD versus *LDLR*^−/−^-RD and Wt-RD or ** *p* < 0.01 HFD groups versus RD groups, as analyzed by two-way ANOVA. (**B**,**C**) higher triglyceride levels were observed in *LDLR*^−/−^-HFD, whereas heavier livers were detected in groups fed with HFD. Data are representative of 7–12 mice per group. Statistical differences were detected by one-way ANOVA for independent measures followed by Tuckey-b or Tamhane test as required. Statistical significance between *LDLR*^−/−^-HFD versus rest of the groups where ** *p* < 0.001 *LDLR*^−/−^-HFD versus rest of the groups and ## *p* < 0.001 significative differences between groups fed with RD versus HFD.

**Figure 2 ijms-24-08042-f002:**
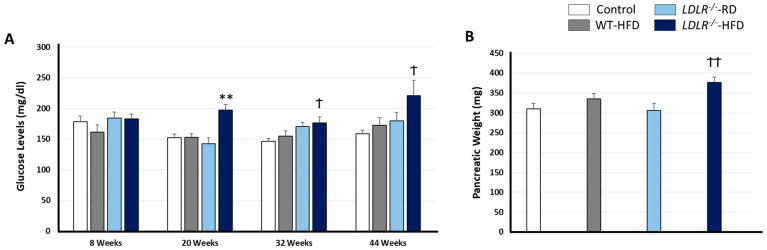
HFD influences plasmatic glucose levels and pancreatic weight in al LDLR^−/−^ background: glucose levels (**A**) and Pancreatic Weights (**B**). Data are representative of 9–12 mice per group. Statistical differences were detected by one-way ANOVA for independent measures followed by Tuckey-b or Tamhane test as required. ** *p* < 0.001 vs. rest of the groups; ^†^ *p* < 0.05 vs. RD groups; ^††^ *p* < 0.001 vs. rest of the groups.

**Figure 3 ijms-24-08042-f003:**
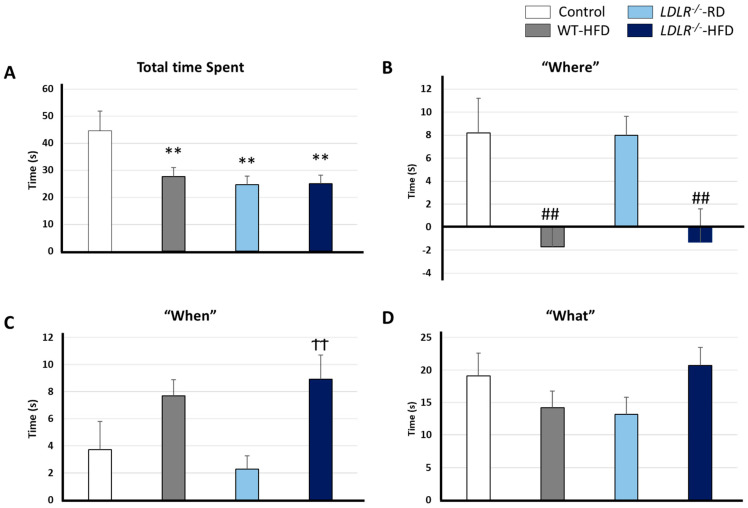
Influence of HFD and LDLR status on episodic memory. The NOD task was carried out in the 43rd week. (**A**) Total time spent exploring the objects. (**B**), (**C**) and (**D**) Paradigms “where”, “when”, and “what”, respectively. Data are representative of 9–12 mice per group. Statistical differences were detected by 1-way ANOVA for independent samples followed by Tukey b. ** *p* < 0.001 vs. control group; ## *p* < 0.001 vs. control and Wt-RD groups; ^††^ *p* < 0.001 vs. control and Wt-RD groups.

**Figure 4 ijms-24-08042-f004:**
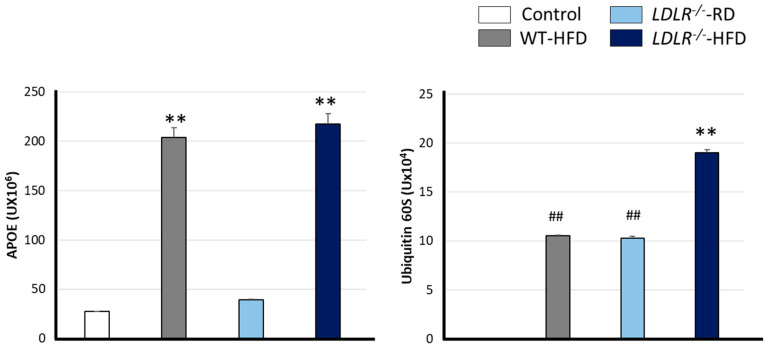
Serum levels of APOE and Ubiquitin 60S. Increased APOE levels were observed in HFD-fed mice. Increased levels of ubiquitin 60S were detected in WT-HFD and LDLR^−/−^-RD compared with the control group, whereas LDLR^−/−^-HFD showed significative higher levels than the rest of the groups. Data are representative of 3 mice per group. Statistical differences were detected by one-way ANOVA for independent measures followed by Tuckey-b. ** *p* < 0.001 vs. rest of the groups; ## *p* < 0.001 vs. control.

**Figure 5 ijms-24-08042-f005:**
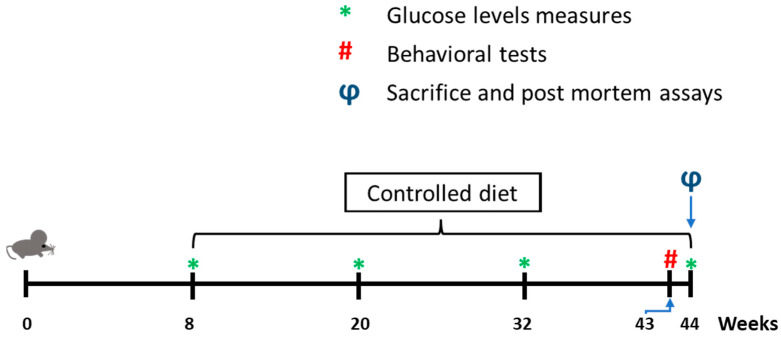
The experimental timeline includes all the processes performed.

**Figure 6 ijms-24-08042-f006:**
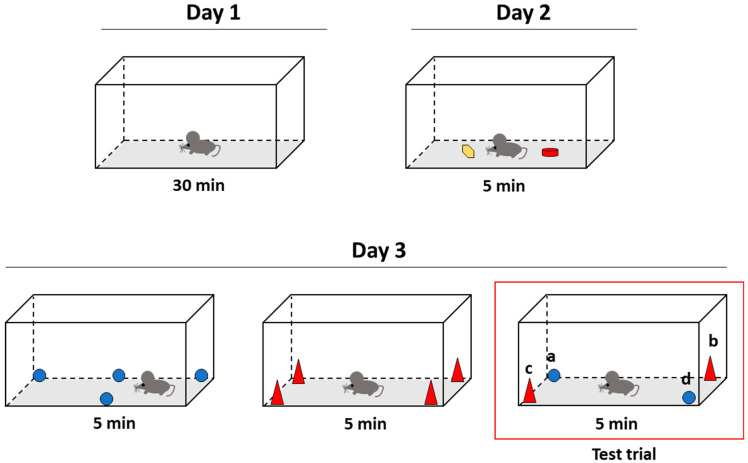
Representative images of the experimental schedule followed for the NOD test [83].

**Table 1 ijms-24-08042-t001:** Gender distribution and experimental groups in the present study.

Groups	Female (*n*)	Male (*n*)	Total (*n*)
Control	4	5	9
Wt-RD	5	4	9
*LDLR*^−/−^-RD	5	7	12
*LDLR*^−/−^-HFD	5	6	11

**Table 2 ijms-24-08042-t002:** Composition and energy value of food intake.

	Regular Diet			High Fat Diet		
	Weight (%)	Kcal/g	Kcal/100 Kcal Intake	Weight (%)	Kcal/g	Kcal/100 Kcal Intake
Protein	20	0.8	24.2	14.8	0.6	12.1
Carbohydrates	77	2.38	72.1	58.85	1.76	35.4
Fat	3	0.12	3.7	29	2.61	52.5

**Table 3 ijms-24-08042-t003:** Mathematical calculations for episodic memory paradigms where a, b, c and d indicate time spent exploring the four corners of the box according to Figure 5 in the test trial.

Paradigm	Calculation
“What”	(a + d) − (b + c)
“Where”	d − a
“When”	a − [(b + c)/2]

## Data Availability

Not applicable.

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
