# Peer review of "Metabolic Disturbance of High-Saturated Fatty Acid Diet in Cognitive Preservation"

_ijms, 2023, doi:10.3390/ijms24098042_

Round 1

Reviewer 1 Report

In the introduction (line 51) authors state “Besides, high cholesterol levels are responsible for many clinical manifestations of 51 obesity (for a review see [15]).”. Could authors provide in the manuscript only few examples of these clinical manifestations?

Is it known if LDLR is express also in microglia?

Could authors specify also in the methods section whether mice used in the experiments are males or females and the “n” value of each experimental group?

How old were mice used in the present study? Do the authors know if the diet could induce different effects according to the age of the mice?

I suggest to include a figure representing a timeline that could give to the readers an idea of when mice were fed and  behavioural tests/ sacrifice were performed.

Could the authors add the statistic also in the figure legends (i.e. *p<0.05 vs...)? I believe this way it would be easier for the reader to understand and interpret data.

In the method section authors state that during the sacrifice also brain were collected, however I was not able to find what they were used for in the paper. Could authors add these experiment/results? Did they look at some neuroinflammatory parameter (i.e. proinflammatory cytokines) in the brain?

Do authors have any idea about the possible effects of this kind of diet on anxiety and depressive like behaviour in mice?

no comments 

Author Response

We would like to thank you for your helpful comments on the manuscript. We are sincerely grateful to you for the time spent reading, commenting, and suggesting changes and modifications, which have greatly improved the quality of the work. We believe that the revisions made based on your reports have significantly improved the original manuscript.

Reviewer 1

In the introduction (line 51) authors state “Besides, high cholesterol levels are responsible for many clinical manifestations of 51 obesity (for a review see [15]).”. Could authors provide in the manuscript only few examples of these clinical manifestations?

Response:  Thanks for the comment, we have restructured this paragraph with your recommendations:

Besides, high cholesterol levels are responsible for many clinical manifestations of obesity as: 1) insulin resistance by glucose transport disruption, 2) dysregulation of transcriptional factors, including a sterol regulatory element-binding proteins or 3) adipocytes hypertrophied promoting obesity development  (for a review see [15]). “

Is it known if LDLR is express also in microglia?

Response: thanks for the comments, as we have included in the discussion, we know that microglia in LDLR mice:

The molecular mechanisms involved here remains elusive but likely involves Aβ pathology and neurotoxicity as previously reported in LDLR-/- raised with hypercaloric diet [77]. However, some studies indicate that LDLR-/- mice is associated with suppressed microglial activation in cortex and hippocampus [71,78]. So we cannot rule out that brain inflammation is decreased in our model and altered cognitive function was altered by others pathways like neurotoxicity or Aβ pathology increased.”

 Could authors specify also in the methods section whether mice used in the experiments are males or females and the “n” value of each experimental group?

Response: Thanks for your comment. we have added an explanatory sentence with the numbers of animals and gender used in each experiment in section 2.5. Statistical Analysis:

“The numbers of male and female mice used for statistical analysis were the specified in table 2, except for the statistics of APOE and Ubiquitin 60S levels in plasma, which was carried out in 3 mice per group (2 males and 1 female)”

How old were mice used in the present study?

Response: Thanks for your comment. Our animals were sacrificed at 44 weeks of age. Mice were on controlled diets from 8 to 44 weeks (36 weeks on diet). We have included Figure 1 with the experimental timeline.

Do the authors know if the diet could induce different effects according to the age of the mice?

Response: Thanks for your comment. In our study we have monitored the evolution of glucose levels, observing how these levels evolve in the combined model. To the best of our knowledge, other authors have applied high-fat diets to this model, but our study is the one that has used long-lived animals under HFD. As we have discussed, shorter times under a hypercaloric diet in LDLR mice are enough to observe alterations such as an increase in beta-amyloid pathology or neurotoxicity.

I suggest to include a figure representing a timeline that could give to the readers an idea of when mice were fed and  behavioural tests/ sacrifice were performed.

Response: Thank for your great suggestion. We have added a new figure 1 with the timeline of the experimental procedures performed

 Could the authors add the statistic also in the figure legends (i.e. *p<0.05 vs...)? I believe this way it would be easier for the reader to understand and interpret data.

Response: Thank for your suggestion, we have added this information in each figure description

In the method section authors state that during the sacrifice also brain were collected, however I was not able to find what they were used for in the paper. Could authors add these experiment/results? Did they look at some neuroinflammatory parameter (i.e. proinflammatory cytokines) in the brain?

Response: Thank you for your appreciation on this point. After the sacrifice of the mice, we weighed their brains and found no significant differences between any of the groups. On the other hand, as we have already commented, LDLR-/- animals show a diminished or limited inflammatory process, it is for this reason that we ruled out exploring the inflammatory process to explain the cognitive deterioration observed.

We have rewritten the 10th paragraph of discussion: “Episodic memory results affected in the first stages of dementia, when brain atrophy is not detectable [68-70]. In the present paper, we have used the NOD task, which allows a rigour exploration of episodic memory. LDLR-/--HFD animals showed worse cognitive efficiency as compared with the rest of the groups, in agreement with previous studies [71,72]. However, the brain weights of all groups were similar, with no signs of brain atrophy being detected (data not shown)…”

Do authors have any idea about the possible effects of this kind of diet on anxiety and depressive like behaviour in mice?

Response: thanks for your interesting question. In figure  5A we can see that the control animals spend more time exploring the objects than the rest of the groups. The literature maintains that a reduction in exploration time may be related to higher levels of stress. However, prior to cognition studies, motor function integrity was assessed in all groups under study. The actimetry test revealed that distance traveled was similar in all the groups supporting  no anxiety or stress.

We have rewritten the 3rd paragraph of “3.3. Cognitive task: NOD test”: “Regarding the former (Figure 4 A), our data clearly shows that Control group spent more exploring time as compared with the rest of the groups [ F(3,34)=4,657 **p=0,008]. However, prior to cognition studies, motor function integrity was assessed in all groups under study. The actimetry test (day 1 of NOD task) revealed that distance and distribution areas traveled was similar in all the groups supporting a no anxiety or stress (data not shown).

Reviewer 2 Report

In this study hyperlipidemia was induced in LDLR-/- mice after feeding with high-saturated fatty acid diet (HFD) for 44 weeks. LDLR-/- -HFD model exhibited obesity, hypertriglyceridemia, higher levels of glucose and early hepatic steatosis. In addition, HFD induced plasmatic APOE increase and ubiquitin 60S levels. These  proteins relate to neuronal integrity and health maintenance. They have detected cognitive dysfunction in LDLR-/- -HFD mice whereas wild type mice fed with HFD preserved their cognition status. The study suggested that central nervous system disfunction is associated with well-established metabolic syndrome. As a consequence, metabolic syndrome boost many behavioral and pathological alterations recognized in dementia, supporting that the control of metabolic parameters could improve the cognitive preservation and prognosis.

The methods followed in this study are well acceptable. The results were well presented and discussed without any ambiguity. In my opinion the manuscript may be accepted for publication with the following minor clarifications.

1.      In Table 1. Composition and energy value of food intake

The protein and carbohydrate   with respect to Weight (%)  and Kcal/g between Regular Diet and High Fat Diet are in the ratio (1: 0.75) but whereas the Caloric intake (%) becomes 50%.  Kindly clarify.

2.      Previous reports showed that WT mice supplied with HFD didn´t present cognitive impairment. Surprisingly, HFD improved working memory and reduced anxiety  in C57BL/6J mice showing that hypercaloric diets might be beneficial in a restricted context.-  You can comment more on this.

3.      Regarding to episodic memory, Contrary to expected no differences were found for the paradigm “what” [F(3,33)=1,714 p=0,183] (Figure 4D). Taken together, we demonstrate that HFD interferes with episodic memory, especially in LDL-/- background.

                 Then how we can say that HFD interferes with episodic memory.

4.      Justify the statement “Our data suggest that a hypercaloric diet is insufficient to promote elevated glycaemic index” but at the same time you indicate that the data support that moderate saturated fatty acid consumption is crucial for chronic proinflammatory diseases prevention as well as healthy cognitive state.

Author Response

We would like to thank you for your helpful comments on the manuscript. We are sincerely grateful to you for the time spent reading, commenting, and suggesting changes and modifications, which have greatly improved the quality of the work. We believe that the revisions made based on your reports have significantly improved the original manuscript.

In this study hyperlipidemia was induced in LDLR-/- mice after feeding with high-saturated fatty acid diet (HFD) for 44 weeks. LDLR-/- -HFD model exhibited obesity, hypertriglyceridemia, higher levels of glucose and early hepatic steatosis. In addition, HFD induced plasmatic APOE increase and ubiquitin 60S levels. These  proteins relate to neuronal integrity and health maintenance. They have detected cognitive dysfunction in LDLR-/- -HFD mice whereas wild type mice fed with HFD preserved their cognition status. The study suggested that central nervous system disfunction is associated with well-established metabolic syndrome. As a consequence, metabolic syndrome boost many behavioral and pathological alterations recognized in dementia, supporting that the control of metabolic parameters could improve the cognitive preservation and prognosis.

The methods followed in this study are well acceptable. The results were well presented and discussed without any ambiguity. In my opinion the manuscript may be accepted for publication with the following minor clarifications.

Response: Thank you very much for taking the time to review our work. We appreciate your assessments and advice to improve this manuscript.

  1. In Table 1. Composition and energy value of food intake

The protein and carbohydrate   with respect to Weight (%)  and Kcal/g between Regular Diet and High Fat Diet are in the ratio (1: 0.75) but whereas the Caloric intake (%) becomes 50%.  Kindly clarify.

Response: Thanks for your comment. We have tried to clarify the meaning of the represented values, specifically, we try to expose the amount of Kcal of each substrate (fats, carbohydrates and proteins) for every 100 Kcal consumed. for this we have renamed the column title that would now be called: "Kcal/100 Kcal intake"

  1. Previous reports showed that WT mice supplied with HFD didn´t present cognitive impairment. Surprisingly, HFD improved working memory and reduced anxiety  in C57BL/6J mice showing that hypercaloric diets might be beneficial in a restricted context.-  You can comment more on this.

Response: Thanks for your recommendation, as we suggest this data are only for restricted research, in order to explain it, we have added these sentences in the paragraph:

“Previous reports showed that WT mice supplied with HFD didn´t present cognitive impairment [34,43]. Surprisingly, HFD improved working memory and reduced anxiety in C57BL/6J mice showing that hypercaloric diets might be beneficial in a restricted context [43]. According to Yoshizaki K., et al. the hypercaloric diet for 7 weeks induced hyperloco-motion, suggesting a lower state of anxiety and a better resolution of the maze-Y test. However, these data must be taken with caution, since the period under HFD was very limited (7 weeks) and the animals were much younger (15 weeks) compared to those we have used in the present study (36 weeks under HFD and 44 weeks of age).”

  1. Regarding to episodic memory, Contrary to expected no differences were found for the paradigm “what” [F(3,33)=1,714 p=0,183] (Figure 4D). Taken together, we demonstrate that HFD interferes with episodic memory, especially in LDL-/- background.

                 Then how we can say that HFD interferes with episodic memory.

Response: thanks for your appreciation. Our data suggest that HFD for 36 weeks was sufficient to generate alterations in episodic memory for the "where" paradigm. In addition, in the LDLR-/- genetic background, it was observed that HFD generated alterations not only in the "where" paradigm, but also in the "when" paradigm. It is true that previous studies with hypercaloric diets did not affect memory, but it should be noted that to the best of our knowledge, studies with HFD as long as ours (36 weeks) have not been carried out. It is feasible that the prolongation of the period under HFD could have caused the limited cognitive impairment described.

In order to clarify this point, we have added these sentences:

“ Regarding to episodic memory, our data point to clear alterations after hypercaloric chronic diets. Indeed, HFD groups showed significant differences as compared with regu-lar diet groups for the paradigm “where” [F(3,36)=5,534, ##p=0.003) (Figure 45B). Further-more, differences were found between LDL-/--HFD versus Control group (F(3,32)=4,793, ††p=0.007) for the paradigm “when” (Figure 54C). Contrary to expected no differences were found for the paradigm “what” [F(3,33)=1,714 p=0,183] (Figure 54D). Previous studies with hypercaloric diets did not describe memory disruption in wild type mice, but it should be noted that to the best of our knowledge, studies with HFD as long as ours (36 weeks) have not been carried out. It is feasible that the prolongation of the period under HFD could have caused the limited cognitive impairment described here in wild type mice.”

  1. Justify the statement “Our data suggest that a hypercaloric diet is insufficient to promote elevated glycaemic index” but at the same time you indicate that the data support that moderate saturated fatty acid consumption is crucial for chronic proinflammatory diseases prevention as well as healthy cognitive state.

Response: We acknowledge the comment of the reviewer. The statement likely indicates that at least part of the cognitive disfunctions observed are due to a glucose independent mechanism and probably inflammation dependent, a good point to explore for future experiments.

 According to our data, we found that glycemic levels were only altered in  a LDLR-/- -HFD context , but not in wild type mice fed with HFD or in LDLR-/- mice fed with regular diet. However, HFD promote a high triglycerides levels and higher body weight. We think lipidic alterations are sufficient to induce a inflammatory process and brain damage, whereas in a next step we can found these metabolic alterations with a glucose disbalance. This new phase could be more aggressive and promote a more severe neurodegeneration processes in our combined model.

Round 2

Reviewer 1 Report

no comments

no comments